# Facile Gram-Scale Synthesis of Co_3_O_4_ Nanocrystal from Spent Lithium Ion Batteries and Its Electrocatalytic Application toward Oxygen Evolution Reaction

**DOI:** 10.3390/nano13010125

**Published:** 2022-12-26

**Authors:** Jaegon Kim, Ho-Geun Kim, Hyun-Su Kim, Cu Dang Van, Min Hyung Lee, Ki-Wan Jeon

**Affiliations:** 1Department of Advanced Technology and Engineering, Graduate School, Silla University, Busan 46958, Republic of Korea; 2Department of Applied Chemistry, Kyung Hee University, Yongin 17104, Republic of Korea

**Keywords:** recycling spent lithium ion battery, Co_3_O_4_ nanoparticle, oxygen evolution reaction, electrocatalyst, alkali leaching

## Abstract

In this study, we demonstrate a new approach to easily prepare spinel Co_3_O_4_ nanoparticles (*s*-Co_3_O_4_ NPs) in the gram-scale from the cathode of spent lithium ion batteries (SLIBs) by the alkali leaching of hexaamminecobalt(III) complex ions. As-obtained intermediate and final products were characterized with powder X-ray diffraction (PXRD), Ultraviolet-Visible (UV–Vis), Fourier transform infrared (FTIR), and Transmission electron microscopy (TEM). Additionally, the synthesized *s*-Co_3_O_4_ NPs showed better electrocatalytic properties toward the oxygen evolution reaction (OER) in comparison to previously reported Co_3_O_4_ NPs and nanowires, which could be due to the more exposed electrocatalytic active sites on the *s*-Co_3_O_4_ NPs. Moreover, the electrocatalytic activity of the *s*-Co_3_O_4_ NPs was comparable to the previously reported RuO_2_ catalysts. By taking advantage of the proposed recycling route, we would expect that various valuable transition metal oxide NPs could be prepared from SLIBs.

## 1. Introduction

Rechargeable lithium-ion batteries (LIBs) have been the most attractive electric power storage system in portable electronic devices such as electric vehicles, mobile phones, and laptops due to their favorable characteristics of lightweight, high energy density, long-life span, etc. [1,2,3]. In particular, in recent years, with rapid growth in the market for electric vehicles, a large quantity of LIBs is required, and a huge amount of SLIBs will also be produced in the near future [4,5,6]. These SLIBs not only contain valuable metal ions (Co, Ni, Li, etc.) with a total metal content ranging from 26 wt% to 76 wt%, but also metal-containing hazardous wastes and organic matter [7,8,9]. If these SLIBs cannot be properly treated, it will give rise to the shortage of valuable metals, especially Co and Li [10,11]. Moreover, it may bring serious health and environmental problems [12,13]. Accordingly, recovering valuable metals from SLIBs would not only be beneficial for the environmental protection, but also helpful for the sustainable supply of strategic resources.

Among the two half-cell electrodes for LIBs, the cathode has been paid more attention for recycling due to a high content of valuable metal ions [14]. Therefore, development of various recycling processes of the cathodes such as hydrometallurgical, pyrometallurgical, and biometallurgical technology have been extensively studied [15,16,17,18,19]. Compared with other methods, much research has paid attention to the hydrometallurgical method due to many advantages such as high efficiency, high metal selectivity, and low energy consumption [20]. Although extensive research of the hydrometallurgical method for recycling SLIBs has been carried out, producing metal oxides through the wet-chemistry route has sparsely been reported so far because the leaching and reducing processes are generally used in the hydrometallurgical approach, which results in generating mostly metals or metal carbonates [21,22]. If transition metal oxides other than metals can be produced through the hydrometallurgical technique, the recycling process and utilization can be facilitated more due to the simple formation and chemical stability of metal oxides in ambient conditions. Among the various transition metal oxides, *s*-Co_3_O_4_ NPs have attracted great attention from both the academic and industrial fields due to its interesting physiochemical properties. Therefore, Co_3_O_4_ NPs have been broadly employed in electrochemical applications, gas sensors, optical and magnetic materials, and catalysts [23,24,25,26,27,28,29,30]. Although SLIBs surprisingly contain a much higher content of Co element than natural ores, very few works synthesizing Co_3_O_4_ NPs from SLIBs have been reported thus far [31,32,33]. Furthermore, the reported investigation required a high temperature calcination and careful control of pH to synthesize micron and nano-sized Co_3_O_4_, respectively. Therefore, the development of other straightforward and facile synthetic methods of Co_3_O_4_ NPs is urgently required for more practical application.

In this study, we report a novel route to synthesize *s*-Co_3_O_4_ NPs from the cathode of SLIB as a Co source through the following processes: (i) refluxing with acetone to remove the binder or organic residues; (ii) leaching Co ions with HCl; (iii) producing Co(NH_3_)_6_·Cl_3_ with NH_4_OH treatment; and (iv) alkali (NaOH) leaching Co(NH_3_)_6_·Cl_3_ to synthesize final product, *s*-Co_3_O_4_ NPs. For this work, LiCoO_2_ was selected as a spent cathode material because it is relatively simple in structure but representative. The synthesized *s*-Co_3_O_4_ NPs showed an average primary particle size of 5.6 ±1.3 nm, as confirmed with TEM. As a demonstration of the practical application of *s*-Co_3_O_4_ obtained from SLIB, the *s*-CO_3_O_4_ was used as an electrocatalyst for the OER. Overpotential of *s*-Co_3_O_4_ NPs for OER was significantly lower than that of the commercially available micron-sized Co_3_O_4_ (*cm*-Co_3_O_4_) and it is comparable to commercially available nano-sized (30 nm) Co_3_O_4_ (*cn*-Co_3_O_4_) NPs. Additionally, the *s*-Co_3_O_4_ NPs showed a similar Tafel slope to that of *cn*-Co_3_O_4_ NPs and a smaller Tafel slope in comparison to the *cm*-Co_3_O_4_. We would like to note that the proposed recycling process of SLIBs in this study can generally be applied to other transition metals contained in SLIBs to synthesize their corresponding transition metal oxide NPs.

## 2. Materials and Methods

### 2.1. Synthesis of s-Co_3_O_4_ NPs from SLIBs

The synthesis procedure of *s*-Co_3_O_4_ NPs from SLIBs is shown in Figure 1.

In a typical preparation, collected Al meshes coated with LiCoO_2_ were treated with acetone to remove polymer resins through a reflux method. After the removal of Al meshes, the precipitated black powder was washed with acetone by vacuum filtration, followed by drying in an oven at 80 °C to obtain LiCoO_2_. To synthesize the Co(NH_3_)_6_·Cl_3_ complex compound, first, 24 g of the LiCoO_2_ black powder was dissolved in 200 mL of 6 *M* HCl solution (Matsunoen Chemicals, Osaka, Japan) and the solution was filtered to eliminate undissolved elements (e.g., residual polymer resins) that may remain. Subsequently, 14 g of activated charcoal and 150 mL of ammonia solution (25–30%, Daejung Chemicals, Siheung, Republic of Korea) were added to the filtered solution and then O_2_ gas bubbling at room temperature, which was to keep the oxidation state of Co^3+^ ions until the color of the solution became an orange color. Afterward, the temperature of the orange colored solution was increased to 70 °C, followed by the activated charcoal subsequently being removed from the heated solution by vacuum filtration. An Erlenmeyer flask containing the filtered solution was placed into an ice bath to facilitate the crystallization of the Co(NH_3_)_6_·Cl_3_ complex compound, which was finally precipitated. Finally, the collected orange colored powder was washed with ethanol (Samchun Chemicals, Seoul, Republic of Korea) and dried in a drying oven at 60 °C for use. In order to synthesize *s*-Co_3_O_4_ NPs, 12 g of the prepared Co(NH_3_)_6_·Cl_3_ powder was dissolved in 200 mL of deionized water at 80 °C, and then 8 g of NaOH (Samchun Chemicals, Seoul, Korea) was slowly added to the heated solution, which resulted in a black precipitate. Finally, the black precipitate was rinsed with deionized water by vacuum filtration to obtain the final product, *s*-Co_3_O_4_ NPs, and it was dried in a drying oven for characterization.

### 2.2. Material Characterization

Structural characterization was performed with a powder diffractometer (XRD-6000, Shimadzu, Kyoto, Japan, Cu-Kα radiation). The FTIR measurement was carried out on a FTIR spectrometer (VERTEX70, Bruker, Billerica, MA, USA). UV–Vis spectroscopy was conducted with UV–Vis spectrophotometer (UV-2600, Shimadzu, Kyoto, Japan). TEM imaging was conducted with a TEM (JEM-2100, JEOL, Tokyo, Japan, acceleration voltage at 200 kV).

### 2.3. Electrochemical Characterization

The electrocatalyst solution for OER was prepared according to the procedure described in the previous report [34]. In a typical preparation, 3 mg of *s*-Co_3_O_4_ NPs was blended with 1 mg of carbon black in 1 mL of isopropanol. A total of 0.08 mL of Nafion solution (5 wt%, Sigma-Aldrich, Burlington, USA) was then added as a binder to improve the adhesion between the *s*-Co_3_O_4_ NPs and the glassy carbon (GC) working electrode. The mixed solution was ultrasonicated for 30 min to obtain a homogeneous dispersion of the catalyst. To fabricate the catalyst loaded working electrodes, 0.02 mL of the catalyst solution was drop-casted on the GC electrode and dried at 25 °C. The analyses of electrocatalytic performance of the prepared working electrode were performed by a typical three-electrode setup coupled with Pt mesh and mercury/mercury oxide (Hg/HgO) electrode as a counter and a reference electrode, respectively. All of the electrochemical tests were performed in the 1 M KOH (pH = 13.6) electrolyte after stabilizing of working electrodes by 20 cycles of cyclic voltammetry (CV) in the range of 0 V to 0.8 V vs. Hg/HgO at a scan rate of 2 mV/s. Next, CV was measured from 0 V to 1.0 V vs. Hg/HgO at the same scan speed to analyze the intrinsic OER catalytic performances (overpotential and Tafel slope) of the *s*-Co_3_O_4_ NPs. For the benchmarking test, working electrodes composed of *cn-*Co_3_O_4_ (Sigma-Aldrich, Burlington, USA) and *cm*-Co_3_O_4_ (Sigma-Aldrich, Burlington, USA) NPs were also prepared via the same processes used for *s*-Co_3_O_4_. Electrochemical impedance spectroscopy (EIS) spectra were measured using the same experimental condition under a bias of 0.7 V vs. Hg/HgO on a broad frequency range from 0.01 to 100,000 Hz with 0.01 V amplitude, and each physical parameter of the catalysts was obtained by fitting the experimental EIS spectra to an equivalent circuit elements using EC-lab (BioLogic, Seyssinet-Pariset, France). All of the reported electrochemical data were corrected by *iR* compensation and converted into E vs. RHE (*E_RHE_*) using the equation of *E_RHE_ = E_Hg/HgO_ +* 0.0592·pH + 0.098 *− iR_s_,* where *E_Hg/HgO_* indicates the experimentally obtained potential based on the Hg/HgO reference electrode and *R_s_* is the solution resistance measured using the Nyquist plot.

## 3. Results and Discussion

In order to identify the crystal structure of the isolated powder in each step, the obtained product was analyzed by PXRD (Figure 2).

The PXRD pattern of the black powder after refluxing treatment in acetone, as shown in Figure 2a, matched the single phase of LiCoO_2_ (JCPDS No. 50-0653), which directly indicates that polymer resins or other possible by-products in the cathode of SLIBs were completely removed. Additionally, it was noted that a refluxing treatment can be an efficient step to solely produce LiCoO_2_. The crystal phase of the synthesized orange colored powder after ammonia solution treatment was validated by PXRD in Figure 2b, which was consistent with the reference pattern (JCPDS No. 70-0787) of the Co(NH_3_)_6_·Cl_3_ complex compound. According to the PXRD examination, the obtained LiCoO_2_ and Co(NH_3_)_6_·Cl_3_ complex compound in each step were successfully isolated and synthesized.

In order to further confirm the crystal structure of the synthesized orange colored product, FTIR and UV–Vis spectroscopy were performed. The FTIR spectrum of the synthesized product was well in agreement with the commercially available Co(NH_3_)_6_·Cl_3_ complex compound (purchased from Sigma-Aldrich, Burlington, NJ, USA), as shown in Figure 3. The FTIR spectrum showed five main characteristic absorption peaks including (i) the strong absorption peak at 3200 cm^−1^ corresponding to the symmetric stretching vibration of the N–H bond and physically adsorbed water molecules in air; (ii) weak broad peak around 1630 cm^−1^, which is ascribed to the asymmetric bending vibration mode of the N–H bond; (iii) the two sharp absorption peaks at 1380 cm^−1^ and 1330 cm^−1^, which were assigned to the N–H symmetric bending vibrations; and (iv) the sharp absorption peak at 825 cm^−1^ belonging to the rocking vibration of Co–N bonding. All of the absorption peaks were in good agreement with the previously reported study [35]. The UV–Vis absorption spectra of the synthesized [Co(NH_3_)_6_]^3+^ and commercial [Co(NH_3_)_6_]^3+^ are shown in Figure 4. The UV–Vis spectra of [Co(NH_3_)_6_]^3+^ show the presence of transitions at 337 nm and 473 nm. The two broad absorption bands based on the Tanabe–Sugano diagram were assigned to ^1^A_1g_→^1^T_2g_ (337 nm) and ^1^A_1g_→^1^T_1g_ (473 nm), which was consistent with the previously reported result [36].

In this study, we aimed to recover valuable metal ion (i.e., Co^3+^) from SLIBs and convert it into its corresponding metal oxides of *s*-Co_3_O_4_ NPs. To achieve this goal, we utilized Co(NH_3_)_6_·Cl_3_ as an intermediate complex compound via treatment with NaOH solution, which resulted in a black colored precipitate. To identify the phase of the product, the PXRD pattern of the product was investigated and compared to that of the commercially available Co_3_O_4_ powder (Sigma-Aldrich, Burlington, USA) in Figure 5. All of the (hkl) peaks of the resultant product were matched with Bragg’s peaks of the reference patterns (JCPDS No. 43-1003) and *cm*-Co_3_O_4_, which directly indicates that the produced final product was *s*-Co_3_O_4_. The Bragg’s peaks of the *s*-Co_3_O_4_ showed a relatively larger full width at half maximum (FWHM) value in comparison with those of the *cm*-Co_3_O_4_, which strongly supports that the particle size of the *s*-Co_3_O_4_ is much smaller than that of *cm*-Co_3_O_4_. In order to verify the crystallite size of the *s*-Co_3_O_4_, the Bragg’s peaks of (311), (111), and (400) were used and averaged out by using Debye–Scherrer equation. It turned out that the calculated crystallite size of the *s*-Co_3_O_4_ NPs was 4.63 nm, which was comparable with the TEM results (Figure 6). To confirm the crystallinity and particle size of the *s*-Co_3_O_4_, TEM analyses were carried out for *s*-Co_3_O_4_ (Figure 6**)**. Synthesized *s*-Co_3_O_4_ NPs were well-dispersed and showed a uniform size and shape (Figure 6a,b). As indicated in the lattice-resolved HRTEM image presented in Figure 6b, the estimated interplanar distance of the *s*-Co_3_O_4_ NPs was 0.46 nm, which was consistent with the *d*-spacing (0.462 nm) of the (111) crystal plane of *s*-Co_3_O_4_ as well as the hexagonal shaped morphology of the (111) facet of *s*-Co_3_O_4_ (inset in Figure 6b). This result is in good agreement with the XRD results, as outlined above (Figure 5). To estimate the average particle size of the *s*-Co_3_O_4_ NPs, the sizes of 100 *s*-Co_3_O_4_ NPs were measured. The TEM images revealed that the average particle size of the *s*-Co_3_O_4_ NPs was 5.6 ± 1.3 nm and the particle size distribution closely resembled a Gaussian distribution (Figure 6c).

An alkali leaching reaction of Co(NH_3_)_6_^3+^, described as follows, could be used to synthesize *s*-Co_3_O_4_ NPs:3 Co(NH_3_)_6_^3+^ (*aq*) + 10 OH^−^ (*aq*) → Co_3_O_4_ (*s*) + 18 NH_3_ (*aq*) + 5 H_2_O (*l*) + 1/2 O_2_ (*g*)(1)

As a utilization of transition metal oxides recycled from SLIBs, the *s*-Co_3_O_4_ NPs through the proposed alkali leaching reaction were used to study the electrocatalytic activity for the OER. In short, electrocatalytic performances for OER were measured using a traditional 3-elctrodes setup using Pt mesh, Hg/HgO, and each electrocatalyst (i.e., *s*-Co_3_O_4_, *cm*-Co_3_O_4_, and *cn*-Co_3_O_4_) coated GC as the counter, reference, and working electrode, respectively (see Electrochemical characterization in the Materials and Methods section for detailed information of the electrochemical test setup). CV measurements were performed to study the OER activity and kinetics of *s*-Co_3_O_4_, *cm*-Co_3_O_4_, and *cn*-Co_3_O_4_. As plotted in Figure 7a, *s*-Co_3_O_4_ and *cn*-Co_3_O_4_ NPs clearly showed distinct electrocatalytic activity with respect to *cm*-Co_3_O_4_ for OER. The overpotential of *s*-Co_3_O_4_ and *cn*-Co_3_O_4_ NPs was 339 mV and 345 mV at 10 mA·cm^−2^, respectively, which was a comparable performance to RuO_2_ [33], whereas the overpotential of *cm*-Co_3_O_4_ (micron size) was 347 mV, which was mainly attributed to size effect. Tafel plots were extracted from a linear range of the CV based on the Tafel equation to compare the electrochemical kinetics for OER. In Figure 7b, the Tafel slopes of *s*-Co_3_O_4_ and *cn*-Co_3_O_4_ NPs were 49.4 mV/dec. and 50.6 mV/dec., respectively, which were lower values in comparison to the previously reported Tafel slopes of the Co_3_O_4_ nanowire (57 mV/dec.) and Co_3_O_4_ NPs (61 mV/dec.) [37,38], reflecting the faster kinetic of *s*-Co_3_O_4_ NPs. On the other hand, *cm*-Co_3_O_4_ showed a higher Tafel slope of 67.6 mV/dec. This result apparently demonstrates that the nano-sized Co_3_O_4_ showed a much better electrocatalytic performance than the micron-sized Co_3_O_4_ crystals for the OER, which was attributed to there being more electrocatalytic active sites exposed on the NPs compared to the micron-scale crystals [39].

We further examined the electron transfer kinetics of the *s*-Co_3_O_4_, *cn*-Co_3_O_4_, and *cm*-Co_3_O_4_ catalysts on the GC electrode by extracting the charge transfer resistance (R*_ct_*) of each catalyst from the fitting of Nyquist plots according to an equivalent circuit model (Figure 7c and Table 1). The R*_ct_* values of the *s*-Co_3_O_4_, *cn*-Co_3_O_4_, and *cm*-Co_3_O_4_ catalysts were 11.91 Ω, 16.59 Ω, and 25.21 Ω, respectively, revealing that the *s*-Co_3_O_4_ catalyst exhibited the fastest charge transfer kinetic (Table 1). The EIS further emphasized that the NPs mainly contributed to the improvement in the OER of *s*-Co_3_O_4_. It is worth noting that the primary particle size of the *s*-Co_3_O_4_ catalysts was about 6 nm (Figure 6), which was the smallest crystal size among the used Co_3_O_4_ crystals in this study.

## 4. Conclusions

In this study, we report a new facile synthetic route to prepare *s*-Co_3_O_4_ NPs in gram-scale from SLIBs through the alkali leaching of Co(NH_3_)_6_·Cl_3_ complex compounds. The synthesized products in each step were characterized with PXRD, FTIR, and UV–Vis analyses to confirm the crystal structure. In addition, the synthesized *s*-Co_3_O_4_ NPs through the proposed synthesized approach in this research exhibited superior electrochemical properties toward the OER compared to the previously reported Co_3_O_4_ NPs and Co_3_O_4_ nanowires where the result was comparable to RuO_2_. The enhanced electrocatalytic activity of *s*-Co_3_O_4_ NPs for the OER including a lower overpotential and Tafel slope as well as faster charge transfer kinetics, which could mainly be due to more exposed active sites from NPs compared to micron crystals. By taking advantage of the proposed recovery process from the LiCoO_2_ cathode of SLIBs, we envisage that this strategy could be extended to the synthesis of various valuable metal oxides nanocrystals from SLIBs made of other transition metals. Finally, we also hope that our proposed recovery process could be utilized in various application fields, which especially require metal oxide nanocrystals.

## Figures and Tables

**Figure 1 nanomaterials-13-00125-f001:**
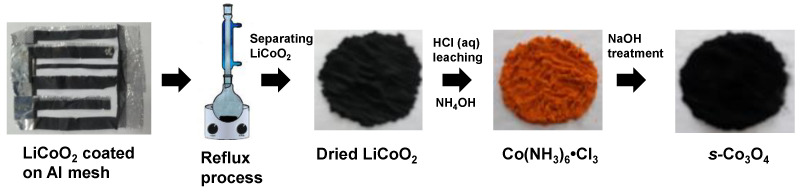
Scheme of the synthesis strategy for *s*-Co_3_O_4_ NPs.

**Figure 2 nanomaterials-13-00125-f002:**
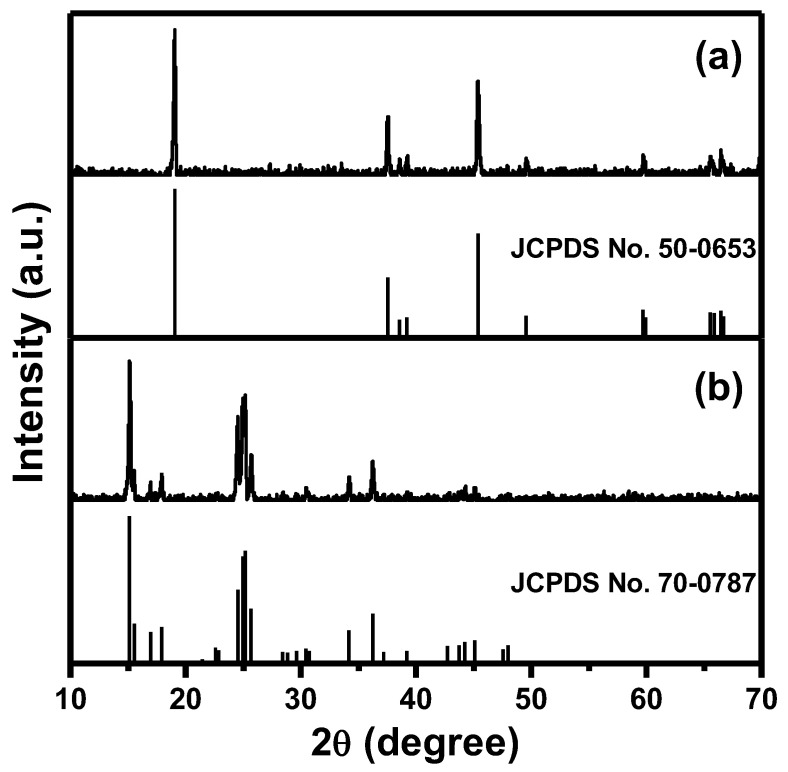
PXRD patterns of (**a**) the obtained LiCoO_2_ after the refluxing process in Figure 1, (**b**) the synthesized Co(NH_3_)_6_·Cl_3_ after the ammonia solution treatment in Figure 1.

**Figure 3 nanomaterials-13-00125-f003:**
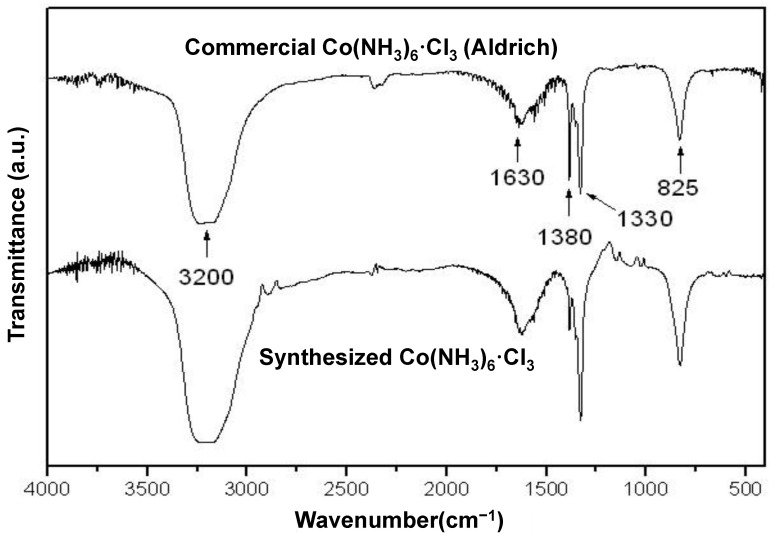
FTIR spectra comparison between the commercially available hexaamminecobalt(III) chloride (top) and synthesized hexaamminecobalt(III) chloride (bottom).

**Figure 4 nanomaterials-13-00125-f004:**
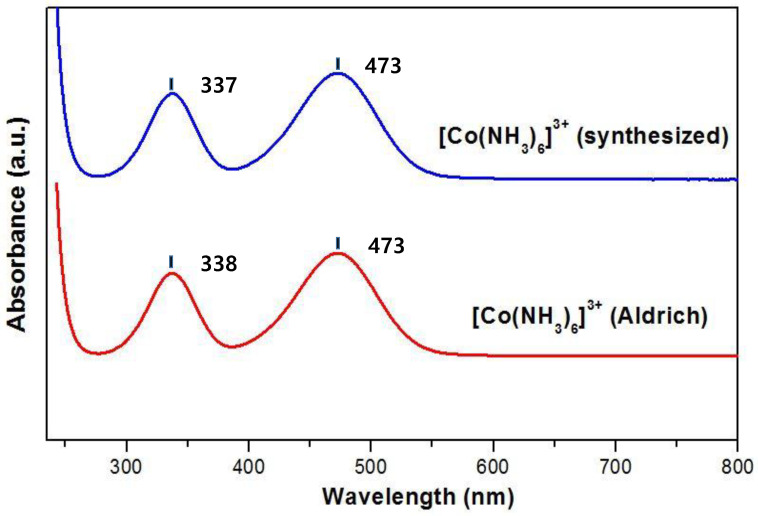
UV–Vis absorption comparison between the commercially available hexaamminecobalt(III) (red) and synthesized hexaamminecobalt(III) cation (blue).

**Figure 5 nanomaterials-13-00125-f005:**
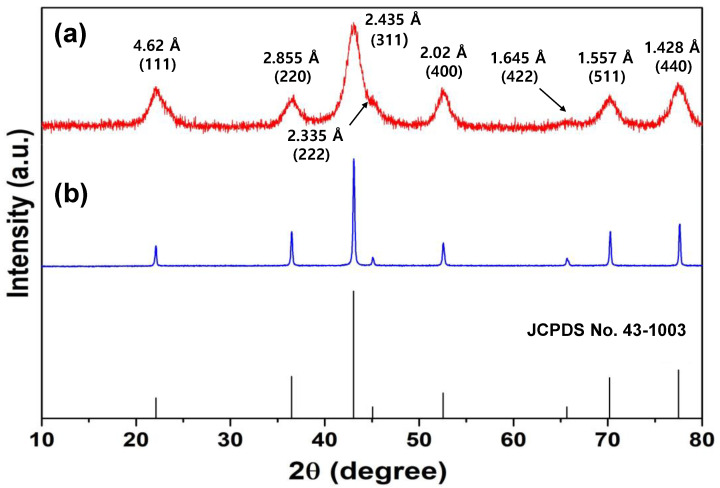
XRD patterns of (**a**) *s*-Co_3_O_4_ after the alkali leaching process and (**b**) *cm*-Co_3_O_4_.

**Figure 6 nanomaterials-13-00125-f006:**
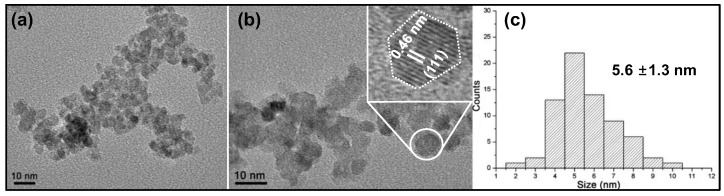
TEM images with (**a**) low magnification and (**b**) high magnification. The high-resolution TEM image of (**b**) shows the lattice spacing of *s*-CO_3_O_4_ NPs. (**c**) Size histogram of the s-Co_3_O_4_ NPs based on the TEM analysis.

**Figure 7 nanomaterials-13-00125-f007:**
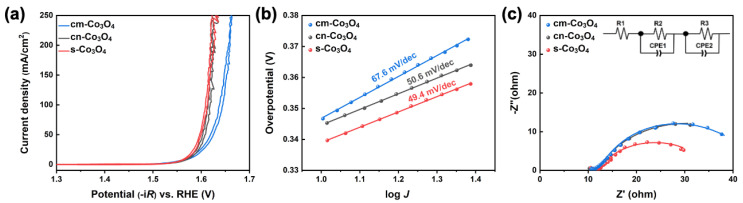
Electrochemical performance of *s*-Co_3_O_4_, *cn*-Co_3_O_4_, and *cm*-Co_3_O_4_ for the OER. (**a**) CV curves collected in 1 M KOH solution at a scan rate of 2 mV/s at room temperature and (**b**) the Tafel plots extracted from (**a**). (**c**) Nyquist plots over a broad frequency range from 0.01 to 100,000 Hz at 0.7 V vs. Hg/HgO with a 0.01 V amplitude.

**Table 1 nanomaterials-13-00125-t001:** The extracted physical parameters of each catalyst from the Nyquist plots in Figure 7c.

Parameter	*cm*-Co_3_O_4_	*cn*-Co_3_O_4_	*s*-Co_3_O_4_
R*_s_* (Ω)	11.25	11.88	11.92
R*_ct_* (Ω)	25.21	16.59	11.91

## Data Availability

The data that support the findings of this study are available from the corresponding authors upon reasonable request.

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
