# Peer review of "Facile Gram-Scale Synthesis of Co3O4 Nanocrystal from Spent Lithium Ion Batteries and Its Electrocatalytic Application toward Oxygen Evolution Reaction"

_nanomaterials, 2022, doi:10.3390/nano13010125_

Round 1

Reviewer 1 Report

The authors report a new approach to prepare spinel Co3O4 nanocrystals in gram scale from spent lithium ion batteries, and the synthesized spinel nanocrystals showed considerable electrocatalytic properties toward OER. After carefully reading this work, I think the over quality of this manuscript needs further improvement. A list of point to be fully addressed is as follows:

1.     There are many spelling and grammar mistakes in the manuscript and supporting information. Please review the full text and correct them. For instance, “5.6+1.3 nm” in line 64, “dried” in line 80, “was” in line 191 and so on.

2.     What is the yield of the synthesized spinel Co3O4 NCs? 

3.     The elemental content of the synthesized spinel Co3O4 NCs should be characterized in detail to accurately analyze the purity.

4.     In Line 227 of Page 7, the authors mentioned, “This result apparently demonstrates that nanosized Co3O4 show much better electrocatalytic performance than micron-sized Co3O4 crystal for OER, which could be more exposed electrocatalytically active sites from the NPs.” Electrochemically surface area (ECSA) and BET surface area should be tested to evaluate the increased number of active sites which is mainly attributed to size effect. Moreover, the intrinsic catalytic activity of the synthesized spinel Co3O4 NCs s should also be calculated and analyzed comparatively.

5.     In Line 237 of Page 8, the authors mentioned, “It is noting that although the solution resistance (Rs) of s-Co3O4 and cn-Co3O4 catalysts are almost the same, the Rct value of s-Co3O4 catalyst is lower than that of cn-Co3O4 catalyst which could be due to more oxygen vacancies or OER-active Co4+ species present on the surface of s- Co3O4 catalyst.”. The oxygen vacancies or the valence state of cobalt need to be further characterized to analyze the origins of the high catalytic activity.

6.     Why the Rct value of s-Co3O4 catalyst is lower than that of cn-Co3O4 catalyst? Please verify the existence of oxygen vacancies and the change of Co valence state by EPR and XPS tests, respectively. Some related works may be helpful for improving the manuscript such as Chemical Engineering Journal, 2022, 441, 136121; Adv. Energy. Mater. 2022, 12, 2202351, Adv. Funct. Mater. 2022, 32, 2207732.

7.     What are the factors that affect the grain size of s-Co3O4? The author should give a detailed discussion.

8.     Will the s-Co3O4 prepared by alkali leaching method produce more defects? The authors should give the necessary explanations.

9.     Is the method reported in this paper applicable to the recover of other transition metal ions?

Reviewer 2 Report

Attached file

Round 2

Reviewer 2 Report

I appreciate that the authors have modified the manuscript according to the comments. However, they do not address properly some of the questions, that are important to be clarified before publication:

Answer to question 8. The authors stated that the measurements in three-electrode cell do not have associated an ohmic drop and this is not true. In fact, they have written in the Fig. 7a “E (-iR)”, which indicates that the ohmic drop (or ohmic losses) are corrected (Kosimaningrum, W. E.; Le, T. X. H.; Holade, Y.; Bechelany, M.; Tingry, S.; Buchari, B.; Noviandri, I.; Innocent, C.; Cretin, M. Surfactant- and Binder-Free Hierarchical Platinum Nanoarrays Directly Grown onto a Carbon Felt Electrode for Efficient Electrocatalysis. ACS Appl. Mater. Interfaces 2017, 9, 22476–22489). Then the authors should amend the text in the experimental section.

Answer to question 9. The authors stated that Co4+ is the reason for the higher performance, but they explain that it will be explained in other future paper. This is not a valid justification, as the mentioned paper is not published and peer reviewed. The authors should give evidence for that (i. e. XPS) or I believe that the hypothesis cannot be supported.

Answer to question 10. First, in the experimental section is written that the applied potential for EIS was 0.7 V vs RHE. If this was the applied potential, the analysis is wrong, as at this potential any HER is occurring. If the applied potential is 1.7 V vs RHE, the results are not coherent, as for this potential in Fig. 7a the performance is superior for the cn- than for s-, while in the EIS plots the Rct is lower for the s- than for cn-. The authors should provide a proper explanation of this fact.

On the other hand, the trend for the OER in the different samples at low current densities is not the same as at high current densities. So, the superiority of s- seems to be limited to a narrow range of overpotentials, while at higher current densities cn- presents lower overpotentials than s-. The authors should mention this difference in the discussion.
